# A Study on Encodings for Neural Architecture Search

**Colin White**
Abacus.AI
San Francisco, CA 94103
colin@abacus.ai

**Willie Neiswanger**
Stanford University and Petuum, Inc.
Stanford, CA 94305
neiswanger@cs.stanford.edu

**Sam Nolen**
Abacus.AI
San Francisco, CA 94103
sam@abacus.ai

**Yash Savani**
Abacus.AI
San Francisco, CA 94103
yash@abacus.ai

## Abstract

Neural architecture search (NAS) has been extensively studied in the past few years. A popular approach is to represent each neural architecture in the search space as a directed acyclic graph (DAG), and then search over all DAGs by encoding the adjacency matrix and list of operations as a set of hyperparameters. Recent work has demonstrated that even small changes to the way each architecture is encoded can have a significant effect on the performance of NAS algorithms [22, 24].

In this work, we present the first formal study on the effect of architecture encodings for NAS, including a theoretical grounding and an empirical study. First we formally define architecture encodings and give a theoretical characterization on the scalability of the encodings we study. Then we identify the main encoding-dependent subroutines which NAS algorithms employ, running experiments to show which encodings work best with each subroutine for many popular algorithms. The experiments act as an ablation study for prior work, disentangling the algorithmic and encoding-based contributions, as well as a guideline for future work. Our results demonstrate that NAS encodings are an important design decision which can have a significant impact on overall performance.[1]

## 1 Introduction

In the past few years, the field of neural architecture search (NAS) has seen a steep rise in interest [2], due to the promise of automatically designing specialized neural architectures for any given problem. Techniques for NAS span evolutionary search, Bayesian optimization, reinforcement learning, gradient-based methods, and neural predictor methods. Many NAS instantiations can be described by the optimization problem $\min_{a \in A} f(a)$, where $A$ denotes a large set of neural architectures, and $f(a)$ denotes the objective function of interest for $a$, which is usually a combination of validation accuracy, latency, or number of parameters. A popular approach is to describe each neural architecture $a$ as a labeled directed acyclic graph (DAG), where each node or edge represents an operation.

Due to the complexity of DAG structures and the large size of the space, neural architecture search is typically a highly non-convex, challenging optimization problem. A natural consideration when designing a NAS algorithm is therefore, *how should we encode the neural architectures to maximize performance?* For example, NAS algorithms may involve manipulating or perturbing architectures, or training a model to predict the accuracy of a given architecture; as a consequence, the representation

of the DAG-based architectures may significantly change the outcome of these subroutines. The majority of prior work has not explicitly considered this question, opting to use a standard encoding consisting of the adjacency matrix of the DAG along with a list of the operations. Two recent papers have shown that even small changes to the architecture encoding can make a substantial difference in the final performance of the NAS algorithm [22, 24]. It is not obvious how to formally define an encoding for NAS, as prior work defines encodings in different ways, inadvertently using encodings which are incompatible with other NAS algorithms.

In this work, we provide the first formal study on NAS encoding schemes, including a theoretical grounding as well as a set of experimental results. We define an encoding as a multi-function from an architecture to a real-valued tensor. We define a number of common encodings from prior work, identifying adjacency matrix-based encodings [26, 24, 21] and path-based encodings [22, 20, 18] as two main paradigms. Adjacency matrix approaches represent the architecture as a list of edges and operations, while path-based approaches represent the architecture as a set of paths from the input to the output. We theoretically characterize the scalability of each encoding by quantifying the information loss from truncation. This characterization is particularly interesting for path-based encodings, which we find to exhibit a phase change at $r^{k/n}$, where $r$ is the number of possible operations, $n$ is the number of nodes, and $k$ is the expected number of edges. In particular, we show that when the size of the path encoding is greater than $r^{2k/n}$, barely any information is lost, but below $r^{k/(2n)}$, nearly all information is lost. We empirically verify these findings.

Next, we identify three major encoding-dependent subroutines used in NAS algorithms: *sample random architecture*, *perturb architecture*, and *train predictor model*. We show which of the encodings perform best for each subroutine by testing each encoding within each subroutine for many popular NAS algorithms. Our experiments retroactively provide an ablation study for prior work by disentangling the algorithmic contributions from the encoding-based contributions. We also test the ability of a neural predictor to generalize to new search spaces, using a given encoding. Finally, for encodings in which multiple architectures can map to the same encoding, we evaluate the average standard deviation of accuracies for the equivalence class of architectures defined by each encoding.

Overall, our results show that NAS encodings are an important design decision which must be taken into account not only at the algorithmic level, but at the subroutine level, and which can have a significant impact on the final performance. Based on our results, we lay out recommendations for which encodings to use within each NAS subroutine. Our experimental results follow the guidelines in the recently released NAS research checklist [9]. In particular, we experiment on two popular NAS benchmark datasets, and we release our code.

**Our contributions.**   We summarize our main contributions below.

- We demonstrate that the choice of encoding is an important, nontrivial question that should be considered not only at the algorithmic level, but at the subroutine level.
- We give a theoretical grounding for NAS encodings, including a characterization of the scalability of each encoding.
- We give an experimental study of architecture encodings for NAS algorithms, disentangling the algorithmic contributions from the encoding-based contributions of prior work, and laying out recommendations for best encodings to use in different settings as guidance for future work.

## 2   Related Work

**Neural architecture search.**   NAS has been studied for at least two decades and has received significant attention in recent years [7, 15, 26]. Some of the most popular techniques for NAS include evolutionary algorithms [11], reinforcement learning [12, 19], Bayesian optimization [6], gradient descent [10], neural predictors [21], and local search [23]. Recent papers have highlighted the need for fair and reproducible NAS comparisons [8, 24, 9]. See the recent survey [2] for more information on NAS research.

**Encoding schemes.**   Most prior NAS work has used the adjacency matrix encoding, [26, 24, 10], which consists of the adjacency matrix together with a list of the operations on each node. A continuous-valued variant has been shown to be more effective for some NAS algorithms [24]. The

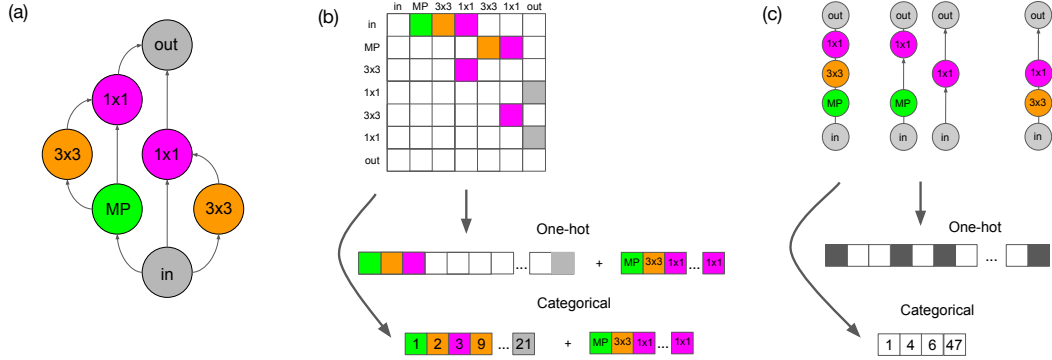

Figure 2.1: (a) An example neural architecture $a$. (b) An adjacency matrix representation of $a$, showing two encodings. (c) A path-based representation of $a$, showing two encodings.

path encoding is a popular choice for neural predictor methods [22, 20, 18], and it was shown that truncating the path encoding leads to a small information loss [22].

Some prior work uses graph convolutional networks (GCN) as a subroutine in NAS [14, 25], which requires retraining for each new dataset or search space. Other work has used intermediate encodings to reduce the complexity of the DAG [16, 4], or added summary statistics to the encoding of feedforward networks [17]. To the best of our knowledge, no paper has conducted a formal study of encodings involving more than two encodings.

# 3   Encodings for NAS

We denote a set of neural architectures $a$ by $A$ (called a search space), and we define an objective function $\ell : A \to \mathbb{R}$, where $\ell(a)$ is typically a combination of the accuracy and the model complexity. We define a neural architecture encoding as an integer $d$ and a multifunction $e : A \to \mathbb{R}^d$ from a set of neural architectures $A$ to a $d$-dimensional Euclidean space $\mathbb{R}^d$, and we define a NAS algorithm $\mathcal{A}$ as a procedure which takes as input a triple $(A, \ell, e)$, and outputs an architecture $a$, with the goal that $\ell(a)$ is as close to $\max_{a \in A} \ell(a)$ as possible. Based on this definition, we consider an encoding $e$ to be a fixed transformation, independent of $\ell$. In particular, NAS components that use $\ell$ to learn a transformation of an input architecture such as graph convolutional networks (GCN) or variational autoencoders (VAE), are considered part of the NAS algorithm rather than the encoding. This is consistent with prior definitions of encodings [18, 24]. However, we do still experimentally compare the fixed encodings with GCNs and VAEs in Section 4.

We define eight encodings split into two popular paradigms: adjacency matrix-based and path-based encodings. We assume that each architecture is represented by a DAG with at most $n$ nodes, at most $k$ edges, at most $P$ paths from input to output, and $q$ choices of operations on each node. We focus on the case where nodes represent operations, though our analysis extends similarly to formulations where edges represent operations. Most of the following encodings have been defined in prior work [24, 22, 18], and we will see in the next section that each encoding is useful for some part of the NAS pipeline.

**Adjacency matrix encodings.**   We first consider a class of encodings that are based on representations of the adjacency matrix. These are the most common types of encodings used in current NAS research.

- The *one-hot adjacency matrix encoding* is created by row-major vectorizing (i.e. flattening) the architecture adjacency matrix and concatenating it with a list of node operation labels. Each position in the operation list is a single integer-valued feature, where each operation is denoted by a different integer. The total dimension is $n(n-1)/2 + n$. See Figure 2.1.

- In the *categorical adjacency matrix encoding*, the adjacency matrix is first flattened (similar to the one-hot encoding described previously), and is then defined as a list of the indices each of which specifies one of the $n(n-1)/2$ possible edges in the adjacency matrix. To

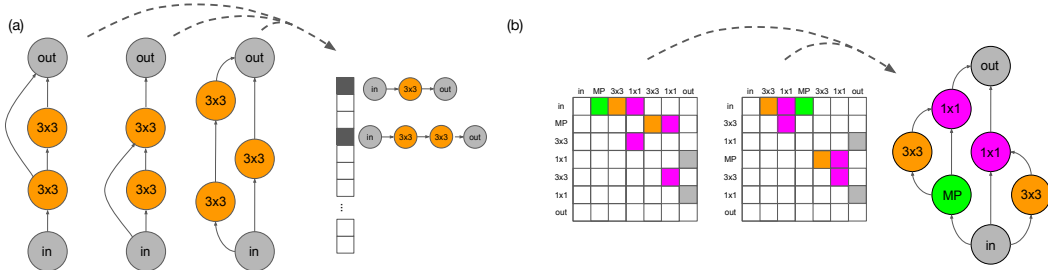

Figure 3.1: (a) An example of three architectures that map to the same path encoding. (b) An example of two adjacency matrices that map to the same architecture.

ensure a fixed length encoding, each architecture is represented by $k$ features, where $k$ is the maximum number of possible edges. We again concatenate this representation with a list of operations, yielding a total dimensionality of $k + n$. See Figure 2.1.

- Finally, the *continuous adjacency matrix encoding* is similar to the one-hot encoding, but each of the features for each edge can take on any real value in $[0, 1]$, rather than just $\{0, 1\}$. We also add a feature representing the number of edges, $1 \le K \le k$. The list of operations is encoded the same way as before. The architecture is created by choosing the $K$ edges with the largest continuous features. The dimension is $n(n - 1)/2 + n + 1$.

The disadvantage of adjacency matrix-based encodings is that nodes are arbitrarily assigned indices in the matrix, which means one architecture can have many different representations (in other words, $e^{-1}$ is not one-to-one). See Figure 3.1 (b).

**Path-based encodings.** Path-based encodings are representations of a neural architecture that are based on the set of paths from input to output that are present within the architecture DAG.

- The *one-hot path encoding* is created by giving a binary feature to each possible path from the input node to the output node in the DAG (for example: `input–conv1x1–maxpool3x3–output`). See Figure 2.1. The total dimension is $\sum_{i=0}^{n} q^i = (q^{n+1} - 1)/(q - 1)$. The *truncated one-hot path encoding*, simply truncates this encoding to only include paths of length $x$. The new dimension is $\sum_{i=0}^{x} q^i$.

- The *categorical path encoding*, is defined as a list of indices each of which specifies one of the $\sum_{i=0}^{n} q^i$ possible paths. See Figure 2.1.

- The *continuous path encoding* consists of a real-valued feature $[0, 1]$ for each potential path, as well as a feature representing the number of paths. Just like the one-hot path encoding, the continuous path encoding can be truncated.

Path-based encodings have the advantage that nodes are not arbitrarily assigned indices, and also that isomorphisms are automatically mapped to the same encoding. Path-based encodings have the disadvantage that different architectures can map to the same encoding ($e$ is not one-to-one). See Figure 3.1 (c).

### 3.1 The scalability of encodings

In this section, we discuss the scalability of the NAS encodings with respect to architecture size. We focus on the one-hot variants of the encodings, but our analysis extends to all encodings. We show that the path encoding can be truncated significantly while maintaining its performance, while the adjacency matrix cannot be truncated at all without sacrificing performance, and we back up our theoretical results with experimental observations in the next section. In prior work, the one-hot path encoding has been shown to be effective on smaller benchmark NAS datasets [20, 22], but it has been questioned whether its exponential $\Theta(q^n)$ length allows it to perform well on very large search spaces [18]. However, a counter-arguement is as follows. The vast majority of features correspond to single line paths using the full set of nodes. This type of architecture is not common during NAS algorithms, nor is it likely to be effective in real applications. Prior work has made the first steps in showing that truncating the path encoding does not harm the performance of NAS algorithms [22].

Consider the popular *sample random architecture* method: given $n$, $r$, and $k \leq \frac{n(n-1)}{2}$, *(1)* choose one of $r$ operations for each node from 1 to $n$; *(2)* for all $i < j$, add an edge from node $i$ to node $j$ with probability $\frac{2k}{n(n-1)}$; *(3)* if there is no path from node 1 to node $n$, goto(1). Given a random graph $G_{n,k,r}$ outputted by this method, let $a_{n,k,\ell}$ denote the expected number of paths from node 1 to node $n$ of length $\ell$ in $G_{n,k,r}$. We define

$$b(k,x) = \frac{\sum_{\ell=1}^{x} a_{n,k,\ell}}{\sum_{\ell=1}^{n} a_{n,k,\ell}}.$$

Given $n < k < n(n-1)/2$ and $0 < x < n$, $b(k,x)$ represents the expected fraction of paths of length at most $x$ in $G_{n,k,r}$. Say that we truncate the path encoding to only include paths of length at most $x$. If $b(k,x)$ is very close to one, then the truncation will result in very little information loss because nearly all paths in a randomly drawn architecture are length at most $x$ with high probability. However, if $b(k,x)$ is bounded away from 1 by some constant, there may not be enough information in the truncated path encoding to effectively run a NAS algorithm.

Prior work has shown that $b(k,x) > 1 - 1/n^2$ when $k < n + O(1)$ and $x > \log n$ [22]. However, no bounds for $b(k,x)$ are known when $k$ is larger than a constant added to $n$. Now we present our main result for the path encoding, which gives a full characterization of $b(k,x)$ up to constant factors. Interestingly, we show that $b(k,x)$ exhibits a phase transition at $x = k/n$. What this means is, for the purposes of NAS, truncating the path encoding to length $r^{k/n}$ contains almost exactly the same information as the full path encoding, and it cannot be truncated any smaller. In particular, if $k \leq n \log n$, the truncated path encoding can be length $n$, which is smaller than the one-hot adjacency matrix encoding. We give the details of the proofs from this section in the full version of this paper.

**Theorem 3.1.** Given $10 \leq n \leq k \leq \frac{n(n-1)}{2}$, and $c > 3$, for $x > 2ec \cdot \frac{k}{n}$, $b(k,x) > 1 - c^{-x+1}$, and for $x < \frac{1}{2ec} \cdot \frac{k}{n}$, $b(k,x) < -2^{\frac{k}{2n}}$.

***Proof sketch.*** Let $G'_{n,k,r}$ denote a random graph after step *(2)* of *sample random architecture*. Then $G'_{n,k,r}$ may not contain a path from node 1 to node $n$. Let $a'_{n,k,\ell}$ denote the expected number of paths of length $\ell$ in $G'_{n,k,r}$. Say that a graph is *valid* if it contains a path from node 1 to node $n$. Then

$$a'_{n,k,\ell} = 0 \cdot (1 - P(G'_{n,k,r} \text{ is valid})) + a_{n,k,\ell} \cdot P(G'_{n,k,r} \text{ is valid}),$$

so $a_{n,k,\ell} = a'_{n,k,\ell}/P(G'_{n,k,r} \text{ is valid})$. Then

$$b(k,x) = \frac{\sum_{\ell=1}^{x} a_{n,k,\ell}}{\sum_{\ell=1}^{n} a_{n,k,\ell}} = \frac{\sum_{\ell=1}^{x} a'_{n,k,\ell}/P(G'_{n,k,r} \text{ is valid})}{\sum_{\ell=1}^{n} a'_{n,k,\ell}/P(G'_{n,k,r} \text{ is valid})} = \frac{\sum_{\ell=1}^{x} a'_{n,k,\ell}}{\sum_{\ell=1}^{n} a'_{n,k,\ell}}.$$

Now we claim $\dfrac{2k}{n(n-1)}\left(\dfrac{2k(n-2)}{(\ell-1)n(n-1)}\right)^{\ell-1} \leq a_{n,k,\ell} \leq \dfrac{2k}{n(n-1)}\left(\dfrac{2ek(n-2)}{(\ell-1)n(n-1)}\right)^{\ell-1}.$

This is because on a path from node 1 to $n$ of length $\ell$, there are $\binom{n-2}{\ell-1}$ choices of intermediate nodes from 1 to $n$. Once the nodes are chosen, we need all $\ell$ edges between the nodes to exist, and each edge exists independently with probability $\frac{2}{n(n-1)} \cdot k$. Then we use the well-known binomial inequalities $\left(\frac{n}{\ell}\right)^{\ell} \leq \binom{n}{\ell} \leq \left(\frac{en}{\ell}\right)^{\ell}$ to finish the claim.

To prove the first part of Theorem 3.1, given $x > 2ec \cdot \frac{k}{n}$, we must upper bound $\sum_{\ell=x+1}^{n} a'_{n,k,\ell}$ and lower bound $\sum_{\ell=1}^{x} a'_{n,k,\ell}$. To lower bound $\sum_{\ell=1}^{x} a'_{n,k,\ell}$, we use $x > 2ec \cdot \frac{k}{n}$ with the claim:

$$\sum_{\ell=x+1}^{n} a_{n,k,\ell} \leq \sum_{\ell=x+1}^{n} \frac{2k}{n(n-1)}\left(\frac{2ek(n-2)}{(\ell-1)n(n-1)}\right)^{\ell-1} \leq \frac{2k}{n(n-1)}\sum_{\ell=x+1}^{n}\left(\frac{1}{c}\right)^{\ell-1}$$

$$\leq \left(\frac{2k}{n(n-1)}\right)\left(\frac{1}{c}\right)^{x-1}$$

We also have $a_{n,k,1} = \frac{2k}{n(n-1)}$ because there is just one path of length 1: the edge from the input node to the output node. Therefore, we have

$$b(k,x) = \frac{\sum_{\ell=1}^{x} a_{n,k,\ell}}{\sum_{\ell=1}^{n} a_{n,k,\ell}} \geq \frac{a_{n,k,1}}{a_{n,k,1} + \sum_{\ell=x+1}^{n} a_{n,k,\ell}} \geq \frac{\frac{2k}{n(n-1)}}{\frac{2k}{n(n-1)} + \left(\frac{2k}{n(n-1)}\right)\left(\frac{1}{c}\right)^{x-1}} \geq 1 - c^{-x+1}.$$

The proof of the second part of Theorem 3.1 uses similar techniques. $\square$

In Figure 4.2, we plot $b(k,x)$ for NASBench-101, which supports Theorem 3.1. Next, we may ask whether the one-hot adjacency matrix encoding can be truncated. However, even removing one bit from the adjacency matrix encoding can be very costly, because each single edge makes the difference between a path from the input node to the output node vs. no path from the input node to the output node. In the next theorem, we show that the probability of a random graph containing any individual edge is at least $2k/(n(n-1))$. Therefore, truncating the adjacency matrix encoding even by a single bit results in significant information loss. In the following theorem, let $E_{n,k,r}$ denote the edge set of $G_{n,k,r}$. Given $1 \leq z \leq n(n-1)/2$, we slightly abuse notation by writing $z \in E_{n,k,r}$ if the edge with index $z$ in the adjacency matrix is in $E_{n,k,r}$.

**Theorem 3.2.** Given $n \leq k \leq \frac{n(n-1)}{2}$ and $1 \leq z \leq n(n-1)/2$, we have $P(z \in E_{n,k,r}) > \frac{2k}{n(n-1)}$.

***Proof.*** Recall that *sample random architecture* adds each edge with probability $2k/(n(n-1))$ and rejects in step *(3)* if there is no path from the input to the output. Define $G'_{n,k,r}$ and *valid* as in the proof of Theorem 3.1 and let $E'_{n,k,r}$ denote the edge set of $G'_{n,k,r}$. Then

$$\frac{P(G'_{n,k,r} \text{ is valid} \mid z \in E'_{n,k,r})}{P(G'_{n,k,r} \text{ is valid})} = \frac{P(z \in E'_{n,k,r} \mid G'_{n,k,r} \text{ is valid})}{P(z \in E'_{n,k,r})} > 1,$$

where the first equality comes from Bayes' theorem, and the inequality follows because there is a natural bijection $\phi$ from graphs with $z$ to graphs without $z$ given by removing $z$, where $G$ is valid if $\phi(G)$ is valid but the reverse does not hold. Therefore,

$$P(z \in E_{n,k,r}) = P(z \in E'_{n,k,r} \mid G'_{n,k,r} \text{ is valid}) = \frac{P(G'_{n,k,r} \text{ is valid} \mid z \in E'_{n,k,r})P(z \in E'_{n,k,r})}{P(G'_{n,k,r} \text{ is valid})}$$

$$> P(z \in E'_{n,k,r}) = \frac{2k}{n(n-1)}. \qquad \square$$

Our theoretical results show that the path encoding can be heavily truncated, while the adjacency matrix cannot be truncated. In the next section, we verify this experimentally (Figure 4.2).

# 4 Experiments

In this section, we present our experimental results. All of our experiments follow the Best Practices for NAS checklist [9]. We discuss our adherence to these practices in the full version of this paper. In particular, we release our code at `https://github.com/naszilla/naszilla`. We run experiments on three search spaces which we describe below.

The NASBench-101 dataset [24] consists of approximately 423,000 neural architectures pretrained on CIFAR-10. The search space is a cell consisting of 7 nodes. The first node is the input, and the last node is the output. The middle five nodes can take one of three choices of operations, and there can be at most 9 edges between the 7 nodes. The NASBench-201 dataset [1] consists of 15625 neural architectures separately trained on each of CIFAR-10, CIFAR-100, and ImageNet16-120. The search space consists of a cell which is a complete directed acyclic graph with 4 nodes. Each edge takes an operation, and there are five possible operations. The DARTS [10] search space is used for large-scale cell-based NAS experiments on CIFAR-10. It contains roughly $10^{18}$ architectures, consisting of two cells: a convolutional cell and a reduction cell, each with six nodes. The first two nodes are input from previous layers, and the last four nodes can take on any DAG structure such that each node has degree two. Each edge can take one of eight operations.

We split up our first set of experiments based on the three encoding-dependent subroutines: *sample random architecture, perturb architecture*, and *train predictor model*. These three subroutines are the only encoding-dependent building blocks necessary for many NAS algorithms.

**Sample random architecture.** Most NAS algorithms use a subroutine to draw an architecture randomly from the search space. Although this operation is more generally parameterized by a distribution over the search space, it is often instantiated with the choice of architecture encoding. Given an encoding, we define a subroutine by sampling each feature uniformly at random. We also compare to sampling each *architecture* uniformly at random from the search space (which does not correspond to any encoding). Note that sampling architectures uniformly at random can be very computationally intensive. It is much easier to sample *features* uniformly at random.

**Perturb architecture.** Another common subroutine in NAS algorithms is to make a small change to a given architecture. The type of modification depends on the encoding. For example, a perturbation might be to change an operation, add or remove an edge, or add or remove a path. Given an encoding and a mutation factor $m$, we define a perturbation subroutine by resampling each feature of the encoding uniformly at random with a fixed probability, so that $m$ features are modified on average.

**Train predictor model.** Many families of NAS algorithms use a subroutine which learns a model based on previously queried architectures. For example, this can take the form of a Gaussian process within Bayesian optimization (BO), or, more recently, a neural predictor model [14, 21, 22]. In the case of a Gaussian process model, the algorithm uses a distance metric defined on pairs of neural architectures, which is typically chosen as the edit distance between architecture encodings [6, 5]. In the case of a neural predictor, the encodings of the queried architectures are used as training data, and the goal is typically to predict the accuracy of unseen architectures.

**Experimental setup and results.** We run multiple experiments for each encoding-dependent subroutine listed above. Many NAS algorithms use more than one subroutine, so in each experiment, we fix the encodings for all subroutines except for the one we are testing. For each NAS subroutine, we experiment on algorithms that depend on the subroutine. In particular, for *random sampling*, we run experiments on the Random Search algorithm. For *perturb architecture*, we run experiments on regularized evolution [13] and local search [23]. For *train predictor model*, we run experiments on BO, testing five encodings that define unique distance functions, as well as NASBOT [6] (which does not correspond to an encoding). We also train a neural predictor model using seven different encodings, as well as GCN [21] and VAE [25]. Since this runs in every iteration of a NAS algorithm [14, 22, 21], we plot the mean absolute error on the test set for different sizes of training data. Finally, we run experiments on BANANAS [22], varying all three subroutines at once. We directly used the open source code for each algorithm, except we changed the hyperparameters based on the encoding, described below. Details on the implementations for each algorithm are discussed in the full version of this paper.

Existing NAS algorithms may have hyperparameters that are optimized for a specific encoding, therefore, we perform hyperparameter tuning for each encoding. We just need to be careful that we do not perform hyperparameter tuning for specific *datasets* (in accordance with NAS best practices [9]). Therefore, we perform the hyperparameter search on CIFAR-100 from NAS-Bench-201, and apply the results on NAS-Bench-101. We defined a search region for each hyperparameter of each algorithm, and then for each encoding, we ran 50 iterations of random search on the full hyperparameter space. We chose the configuration that minimizes the validation loss of the NAS algorithm after 200 queries.

In each experiment, we report the test error of the neural network with the best validation error after time $t$, for $t$ up to 130 TPU hours. We run 300 trials for each algorithm and record the mean test errors. See Figure 4.1 for the results on NASBench-101. We present more experiments for NASBench-201 and the DARTS search space in the full version of this paper, seeing largely the same trends. Depending on the subroutine, two encodings might be functionally equivalent, which is why not all encodings appear in each experiment (for example, in local search, there is no difference between one-hot and categorical encodings). There is no overall best encoding; instead, each encoding has varied performance for each subroutine, and the results in Figure 4.1 act as a guideline for which encodings to use in which subroutines. As a rule of thumb, the adjacency matrix-based encodings perform well for the sample random architecture and perturb architecture subroutines, but the path-based encodings far outperformed the adjacency matrix-based encodings for the train predictor model subroutines.

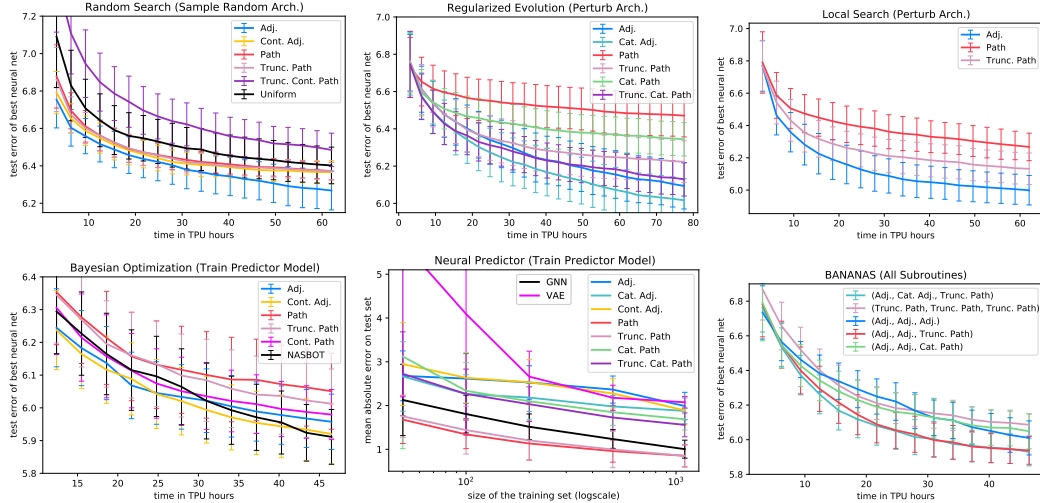

Figure 4.1: Experiments on NASBench-101 with different encodings, keeping all but one subroutine fixed: *random sampling* (top left), *perturb architecture* (top middle, top right), *train predictor model* (bottom left, bottom middle), or varying all three subroutines (bottom right).

Categorical, one-hot, adjacency-based, path-based, and continuous encodings are all best in certain settings. Some of our findings explain the success of prior algorithms, e.g., regularized evolution using the categorical adjacency encoding, and BANANAS using the path encoding in the meta neural network. We also show that combining the best encodings for each subroutine in BANANAS yields the best performance. Finally, we show that the path encoding even outperforms GCNs and VAEs in the neural predictor experiment.

In Figure 4.1, *Trunc. Path* denotes the path encoding truncated from $\sum_{i=0}^{5} 3^i = 364$ to $\sum_{i=0}^{3} 3^i = 40$. As predicted by Theorem 3.1, this does not decrease performance. In fact, in regularized evolution, the truncation improves performance significantly because perturbing with the full path encoding is more likely to add uncommon paths that do not improve accuracy. We also evaluate the effect of truncating the one-hot adjacency matrix encoding on regularized evolution, from the full 31 bits (on NASBench-101) to 0 bits, and the path encoding from 31 bits (out of 364) to 0 bits. See Figure 4.2. The path encoding is much more robust to truncation, consistent with Theorems 3.1 and 3.2.

**Outside search space experiment.** In the set of experiments above, we tested the effect of encodings on a neural predictor model by computing the mean absolute error between the predicted vs. actual errors on the test set, and also by evaluating the performance of BANANAS when changing the encoding of its neural predictor model. The latter experiment tests the predictor model's ability to predict the *best* architectures, not just all architectures on average. We take this one step further and test the ability of the neural predictor to generalize beyond the search space on which it was trained. We set up the experiment as follows. We define the training search space as a subset of NASBench-101: architectures with at most 6 nodes and 7 edges. We define the disjoint test search space as architectures with 6 nodes and 7 to 9 edges. The neural predictor is trained on 1000 architectures and predicts the validation loss of the 5000 architectures from the test search space. We evaluate the losses of the ten architectures with the highest predicted validation loss. We run 200 trials for each encoding and average the results. See Table 1. The adjacency encoding performed the best. An explanation is that for the path encoding, there are features (paths) in architectures from the test set that do not exist in the training set. This is not the case for the adjacency encoding: all features (edges) from architectures in the test set have shown up in the training set.

**Equivalence class experiments.** Recall that the path encoding function $e$ is not one-to-one (see Figure 3.1). In general, this is not desirable because information is lost when two architectures map to the same encoding. However, if the encoding function only maps architectures with similar accuracies to the same encoding, then the behavior is beneficial. On the NASBench-101 dataset, we compute the path encoding of all 423k architectures, and then we compute the average standard

Table 1: Ability of neural predictor with different encodings to generalize beyond the search space.

| Encoding | Validation error | | Test error | |
|---|---|---|---|---|
| | Top 10 avg. | Top 1 avg. | Top 10 avg. | Top 1 avg. |
| Adjacency | **5.888** | **5.505** | **6.454** | **6.056** |
| Categorical Adjacency | 7.589 | 6.191 | 8.155 | 7.086 |
| Path | 5.967 | 5.606 | 6.616 | 6.335 |
| Truncated Path | 6.082 | 5.644 | 6.712 | 6.452 |
| Categorical Path | 6.357 | 5.703 | 6.939 | 6.489 |
| Truncated Categorical Path | 6.339 | 5.895 | 6.918 | 6.766 |

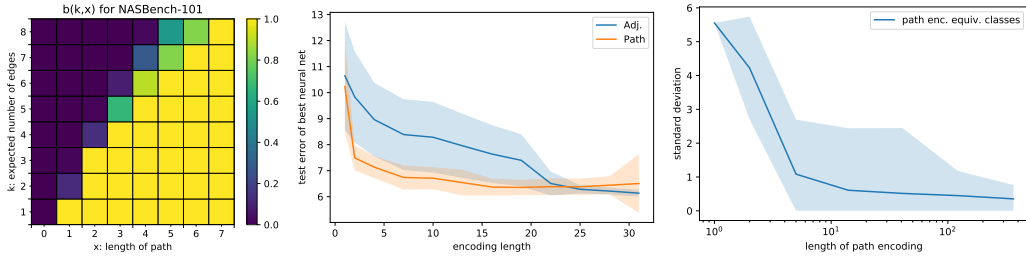

Figure 4.2: Plot of $b(k, x)$ on NASBench-101 (left), which is consistent with Theorem 3.1. Truncation of encodings for regularized evolution on NASBench-101 (middle). Average standard deviation of accuracies within each equivalence class defined by the path encoding at different levels of truncation on NASBench-101 (right).

deviation of accuracies among architectures with the same encoding (i.e., we look at the standard deviations within the equivalence classes defined by the encoding). See Figure 4.2. The result is an average standard deviation of 0.353%, compared to the 5.55% standard deviation over the entire set of architectures.

## 5   Conclusion

In this paper, we give the first formal study of encoding schemes for neural architecture search. We define eight different encodings and characterize the scalability of each one. We then identify three encoding-dependent subroutines used by NAS algorithms—*sample random architecture*, *perturb architecture*, and *train predictor model*—and we run experiments to find the best encoding for each subroutine in many popular algorithms. We also conduct experiments on the ability of a neural predictor to generalize beyond the training search space, given each encoding. Our experimental results allow us to disentangle the algorithmic and encoding-based contributions of prior work, and act as a set of guidelines for which encodings to use in future work. Overall, we show that encodings are an important, nontrivial design decision in the field of NAS. Designing and testing new encodings is an exciting next step.

## 6   Broader Impact

Our work gives a study on encodings for neural architecture search, with the goal of helping future researchers improve their NAS algorithms. Therefore, this work may not have a direct impact on society, since it is two levels of abstraction from real applications, but it can indirectly impact society. As an example, our work may inspire the creation of a new state-of-the-art NAS algorithm, which is then used to improve the performance of various deep learning algorithms, which can have both beneficial and detrimental uses (e.g. optimizers that reduce $CO_2$ emissions, or deep fake generators). Due to the recent push for the AI community to be more conscious and prescient about the societal impact of its work [3], we are hoping that future AI models, including ones influenced by our work, will have a positive impact on society.

# 7 Acknowledgments

We thank the anonymous reviewers for their helpful suggestions. WN was supported by U.S. Department of Energy Office of Science under Contract No. DE-AC02-76SF00515.

## Footnotes

[1]See the full-length paper here: https://arxiv.org/abs/2007.04965. Our code is available at https://github.com/naszilla/naszilla.

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
