[Supplementary Material]

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

. For example, the one-hot adjacency matrix encoding performs well in most settings, but is quite poor in the neural predictor subroutine. Categorical, one-hot, adjacency-based, path-based, and continuous encodings are all best in certain settings. Some of our findings explain the success of prior algorithms, e.g., regularized evolution using the categorical adjacency encoding, and BANANAS using the path encoding in the meta neural network. Some of our results show new discoveries, for example, the continuous adjacency encoding has previously never been used for NAS with BO, and it outperforms all other encodings. We also show that combining the best encodings for each subroutine in BANANAS yields the best performance.

In Figure 4.1, *Trunc. Path* denotes the path encoding truncated from $\sum_{i=0}^{5} 3^i = 364$ to $\sum_{i=0}^{3} 3^i = 40$. As predicted by Theorem 3.1, this does not decrease performance. In fact, in regularized evolution, the truncation improves performance significantly because perturbing with the full path encoding is more likely to add uncommon paths that do not improve accuracy. We also evaluate the effect of truncating the one-hot adjacency matrix encoding on regularized evolution, from the full 31 bits (on NASBench-101) to 0 bits, and the path encoding from 31 bits (out of 364) to 0 bits. See Figure 4.2. The path encoding is much more robust to truncation, consistent with Theorems 3.1 and 3.2.

**Outside search space experiment.** In the set of experiments above, we tested the effect of encodings on a neural predictor model by computing the mean absolute error between the predicted vs. actual errors on the test set, and also by evaluating the performance of BANANAS when changing the encoding of its neural predictor model. The latter experiment tests the predictor model's ability to predict the *best* architectures, not just all architectures on average. We take this one step further and test the ability of the neural predictor to generalize beyond the search space on which it was trained. We set up the experiment as follows. We define the training search space as a subset of NASBench-101: architectures with at most 6 nodes and 7 edges. We define the disjoint test

Table 1: Ability of neural predictor with different encodings to generalize beyond the search space.

| Encoding | Validation error | | Test error | |
|---|---|---|---|---|
| | Top 10 avg. | Top 1 avg. | Top 10 avg. | Top 1 avg. |
| Adjacency | **5.888** | **5.505** | **6.454** | **6.056** |
| Categorical Adjacency | 7.589 | 6.191 | 8.155 | 7.086 |
| Path | 5.967 | 5.606 | 6.616 | 6.335 |
| Truncated Path | 6.082 | 5.644 | 6.712 | 6.452 |
| Categorical Path | 6.357 | 5.703 | 6.939 | 6.489 |
| Truncated Categorical Path | 6.339 | 5.895 | 6.918 | 6.766 |

Figure 4.2: Plot of $b(k, x)$ on NASBench-101 (left). Truncation of encodings for regularized evolution on NASBench-101 (middle). Average standard deviation of accuracies within each equivalence class defined by the path encoding at different levels of truncation on NASBench-101 (right).

search space as architectures with 6 nodes and 7 to 9 edges. The neural predictor is trained on 1000 architectures and predicts the validation loss of the 5000 architectures from the test search space. We evaluate the losses of the ten architectures with the highest predicted validation loss. We run 200 trials for each encoding and average the results. See Table 1. The adjacency encoding performed the best. An explanation is that for the path encoding, there are features (paths) in architectures from the test set that do not exist in the training set. This is not the case for the adjacency encoding: all features (edges) from architectures in the test set have shown up in the training set.

**Equivalence class experiments.** Recall that the path encoding function $e$ is not onto (see Figure 3.1). In general, this is not desirable because information is lost when two architectures map to the same encoding. However, if the encoding function only maps architectures with similar accuracies to the same encoding, then the behavior is beneficial. On the NASBench-101 dataset, we compute the path encoding of all 423k architectures, and then we compute the average standard deviation of accuracies among architectures with the same encoding (i.e., we look at the standard deviations within the equivalence classes defined by the encoding). The result is an average standard deviation of 0.353%, compared to the 5.55% standard deviation over the entire set of architectures. See Figure 4.2.

# 5 Conclusion

In this paper, we give the first formal study of encoding schemes for neural architecture search. We define eight different encodings and characterize the scalability of each one. We then identify three encoding-dependent subroutines used by NAS algorithms, *sample random architecture*, *perturb architecture*, and *train predictor model*, and we run experiments to find the best encoding for each subroutine in many popular algorithms. We also conduct experiments on the ability of a neural predictor to generalize beyond the training search space, given each encoding. Our experimental results allow us to disentangle the algorithmic and encoding-based contributions of prior work, and act as a guideline for the encodings to use in future work. Overall, we show that encodings are an important, nontrivial design decision in the field of NAS. Designing and testing new encodings is an exciting next step.

## 6 Broader Impact

Our work gives a study on encodings for neural architecture search, with the goal of helping future researchers improve their NAS algorithms. Therefore, this work will not directly impact society, since it is two levels of abstraction from real applications, but it can indirectly impact society. As an example, our work may inspire the creation of a new state-of-the-art NAS algorithm, which is then used to improve the performance of several deep learning algorithms, ranging from optimizers that reduce $CO_2$ emissions, to deep fake generators.

Since our work is two levels up the stack, we have much less control over the net impact of our work on society. Due to the recent push for the AI community to be more conscious and clairvoyant about the societal impact of its work, [3] we are cautiously optimistic that our work will have a net positive impact.

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

## A Details from Section 3 (Encodings for NAS)

We give the details from Section 3. We restate the random graph model here more formally.

**Definition A.1.** Given nonzero integers $n, r$, and $k < {}^{n(n-1)}/_2$, a random graph $G_{n,k,r}$ is generated as follows:

(1) Denote $n$ nodes by 1 to $n$ and label each node randomly with one of $r$ operations.

(2) For all $i < j$, add edge $(i, j)$ with probability $\frac{2k}{n(n-1)}$.

(3) If there is no path from node 1 to node $n$, goto (1).

Let $G'_{n,k,r}$ denote the random graph outputted by the above procedure without step (3). Since the number of pairs $(i, j)$ such that $i < j$ is $\frac{n(n-1)}{2}$, the expected number of edges of $G'_{n,k,r}$ is $k$. Define $a_{n,k,\ell}$ as the expected number of paths from node 1 to node $n$ of length $\ell$ in $G'_{n,k,r}$. Formally, we set $\mathcal{P} = \{$paths from node 1 to $n$ in $G'_{n,k,r}\}$, and define

$$a_{n,k,\ell} = \mathbb{E}\left[|p \in \mathcal{P}| \mid |p| = \ell\right].$$

Recall that

$$b(k, x) = \frac{\sum_{\ell=1}^{x} a_{n,k,\ell}}{\sum_{\ell=1}^{n} a_{n,k,\ell}}.$$

In the next theorem, we give a full characterization of $b(k, x)$ in terms of $k$ and $n$, up to constant factors. We prove there exists a phase transition for $b(k, x)$ at $x = \frac{k}{n}$. As noted by prior work [24], there are two caveats when applying this type of theorem to NAS performance. The theorem considers the distribution from Definition A.1, not the distribution of architectures encountered in a real search, and the most common paths in the distribution are not necessarily the ones with the most entropy in predicting whether an architecture has a high accuracy. However, two prior works have experimentally showed that truncating the path encoding does not decrease performance [22, 24], and we gave even more experimental evidence in Section 4.

**Theorem 3.1 (restated).** Given $n \le k \le \frac{n(n-1)}{2}$, and $c > 3$, for $x > 2ec \cdot \frac{k}{n}$, $b(k, x) > 1 - c^{-x+1}$, and for $x < \frac{1}{2ec} \cdot \frac{k}{n}$, $b(k, x) < -2^{\frac{k}{2n}}$.

To prove Theorem 3.1, we use the well-known bounds on binomial coefficients, e.g. [16].

**Theorem A.2.** Given $0 \le \ell \le n$,

$$\left(\frac{n}{\ell}\right)^\ell \le \binom{n}{\ell} \le \left(\frac{en}{\ell}\right)^\ell.$$

Now we give upper and lower bounds on $a_{n,k,\ell}$ which will be used for the rest of the proofs. The next fact is similar to Lemma C.3 from BANANAS [24].

**Fact A.3.** Given $n \le k \le \frac{n(n-1)}{2}$, and $0 < x < n$, we have

$$\frac{2k}{n^2}\left(\frac{2k(1-\epsilon)}{(\ell-1)n}\right)^{\ell-1} \le a_{n,k,\ell} \le \frac{2k}{n^2}\left(\frac{2ek(1-\epsilon)}{(\ell-1)n}\right)^{\ell-1}$$

*Proof.* First, we have

$$a_{n,k,\ell} = \binom{n-2}{\ell-1}\left(\frac{2k}{n(n-1)}\right)^\ell$$

because on a path from node 1 to node $n$ with length $\ell$, there are $\binom{n-2}{\ell-1}$ choices of intermediate nodes from 1 to $n$. Once the nodes are chosen, we need all $\ell$ edges between the nodes to exist, and each edge exists independently with probability $\frac{2}{n(n-1)} \cdot k$. Then we achieve the desired result by applying Theorem A.2. □

Now we prove the upper bound of Theorem 3.1.

**Lemma A.4.** Given $n \le k \le \frac{n(n-1)}{2}$ and $c > 2$, for $x > \frac{2eck}{n}$, $b(k, x) > 1 - c^{-x+1}$.

*Proof.* Given $n \leq k \leq \frac{n(n-1)}{2}$ and $x > \frac{2eck}{n}$, we give a lower bound for $\sum_{\ell=1}^{x} a_{n,k,\ell}$ and an upper bound for $\sum_{\ell=x+1}^{n} a_{n,k,\ell}$.

When $\ell = 1$, we have $\binom{n-2}{\ell-1} = 1$. Therefore,

$$\sum_{\ell=1}^{x} a_{n,k,\ell} \geq a_{n,k,1} = \left(\frac{2k}{n(n-1)}\right) \geq \frac{2k}{n^2}.$$

Now we upper bound $\sum_{\ell=x}^{n} a_{n,k,\ell}$.

$$
\begin{aligned}
\sum_{\ell=x+1}^{n} a_{n,k,\ell} &\leq \sum_{\ell=x+1}^{n} \frac{2k}{n^2} \left(\frac{2ek(1-\epsilon)}{(\ell-1)n}\right)^{\ell-1} \\
&= \frac{2k}{n^2} \sum_{\ell=x+1}^{n} \left(\frac{2ek(1-\epsilon)}{(\ell-1)n}\right)^{\ell-1} \\
&\leq \frac{2k}{n^2} \sum_{\ell=x+1}^{n} \left(\frac{1}{c}\right)^{\ell-1} \\
&= \left(\frac{2k}{n^2}\right) \left(\frac{1}{c}\right)^{x} \sum_{\ell=0}^{\infty} \left(\frac{1}{c}\right)^{\ell} \qquad \text{(A.1)} \\
&= \left(\frac{2k}{n^2}\right) \left(\frac{1}{c}\right)^{x} \left(\frac{c}{c-1}\right) \\
&= \left(\frac{2k}{n^2}\right) \left(\frac{1}{c}\right)^{x-1}
\end{aligned}
$$

In inequality A.1, we use the fact that for all $\ell \geq x$,

$$\ell \geq x > \frac{2eck}{n} \implies \frac{2ek(1-\epsilon)}{n\ell} \leq \frac{1}{c}$$

when $c > 2$, since $1 - \epsilon > \frac{n-2}{n-1}$ in Fact A.3.

Therefore, we have

$$
\begin{aligned}
b(k,x) &= \frac{\sum_{\ell=1}^{x} a_{n,k,\ell}}{\sum_{\ell=1}^{n} a_{n,k,\ell}} \\
&= \frac{\sum_{\ell=1}^{x} a_{n,k,\ell}}{\sum_{\ell=1}^{x} a_{n,k,\ell} + \sum_{\ell=x+1}^{n} a_{n,k,\ell}} \\
&\geq \frac{\frac{2k}{n^2}}{\frac{2k}{n^2} + \left(\frac{2k}{n^2}\right)\left(\frac{1}{c}\right)^{x-1}} \\
&= \frac{1}{1 + \left(\frac{1}{c}\right)^{x-1}} \\
&\geq 1 - c^{-x+1}.
\end{aligned}
$$

$\square$

Now we prove the lower bound for Theorem 3.1.

**Lemma A.5.** *Given $n \leq k \leq \frac{n(n-1)}{2}$ and $c > 3$, for $x < \frac{k}{2ecn}$, $b(k,x) < 2^{-\frac{k}{2n}}$.*

*Proof.* Given $n \leq k \leq \frac{n(n-1)}{2}$ and $x < \frac{k}{2ecn}$, now we give an upper bound for $\sum_{\ell=1}^{x} a_{n,k,\ell}$ and a lower bound for $\sum_{\ell=x+1}^{n} a_{n,k,\ell}$.

428 First we make the following claim. For all $1 \le \ell \le x < \frac{k}{2ecn}$, we have

$$\left(\frac{2ek(1-\epsilon)}{(\ell-1)n}\right)^{\ell} < \left(4e^2 c(1-\epsilon)\right)^{\frac{k}{2ecn}} . \tag{A.2}$$

429 To prove the claim, we have

$$\left(\frac{2ek(1-\epsilon)}{(\ell-1)n}\right)^{\ell} = e^{\log\left(\frac{2ek(1-\epsilon)}{(\ell-1)n}\right)\ell} \tag{A.3}$$

$$= e^{\ell \log \frac{1}{\ell} + \ell \log\left(\frac{2ek(1-\epsilon)}{n}\right)}$$

$$\le e^{\frac{k}{2ecn}\log\left(\frac{2ecn}{k}\right) + \frac{k}{2ecn}\log\left(\frac{2ek(1-\epsilon)}{n}\right)} \tag{A.4}$$

$$= e^{\log\left(4e^2 c(1-\epsilon)\right)\frac{k}{2ecn}} \tag{A.5}$$

$$= \left(4e^2 c(1-\epsilon)\right)^{\frac{k}{2ecn}}$$

430 In inequality A.5, we use the fact that $\frac{k}{(\ell-1)n} < 1$ and $y > \log y$ for all $y > 1$.

431 Then we have

$$\sum_{\ell=1}^{x} a_{n,k,\ell} \le \sum_{\ell=1}^{x} \frac{2k}{n^2} \left(\frac{2ek(1-\epsilon)}{(\ell-1)n}\right)^{\ell-1}$$

$$= \frac{2k}{n^2} \sum_{\ell=1}^{x} \left(\frac{2ek(1-\epsilon)}{(\ell-1)n}\right)^{\ell-1}$$

$$\le \left(\frac{2k}{n^2}\right) \cdot x \cdot \left(4e^2 c(1-\epsilon)\right)^{\frac{k}{2ecn}} .$$

432 Now we give the lower bound for the other side of the summation.

$$\sum_{\ell=x}^{n} a_{n,k,\ell} = \sum_{\ell=x}^{n} \frac{2k}{n^2} \left(\frac{2k(1-\epsilon)}{(\ell-1)n}\right)^{\ell-1} \ge \frac{2k}{n^2} \sum_{\ell=\frac{k}{n}}^{\frac{k}{n}} \left(\frac{2k(1-\epsilon)}{(\ell-1)n}\right)^{\ell-1} = \frac{2k}{n^2} \left(2(1-\epsilon)\right)^{\frac{k}{n}}$$

433 Therefore,

$$b(k,x) = \frac{\sum_{\ell=1}^{x} a_{n,k,\ell}}{\sum_{\ell=1}^{n} a_{n,k,\ell}}$$

$$\le \frac{\sum_{\ell=1}^{x} a_{n,k,\ell}}{\sum_{\ell=x+1}^{n} a_{n,k,\ell}}$$

$$\le \frac{\left(\frac{2k}{n^2}\right) \cdot x \cdot \left(4e^2 c(1-\epsilon)\right)^{\frac{k}{2ecn}}}{\frac{2k}{n^2} \left(2(1-\epsilon)\right)^{\frac{k}{n}}}$$

$$\le x \cdot (2e)^{\frac{k}{ecn}} \left(c\right)^{\frac{k}{2ecn}} \left(2\right)^{-\frac{k}{n}}$$

$$\le x \cdot 2^{-\frac{k}{n}\left(1-\frac{1}{c}-\frac{\log c}{2ec}\right)}$$

$$\le 2^{-\frac{k}{2n}}$$

434 $\qquad\qquad\qquad\qquad\qquad\qquad\qquad\qquad\qquad\qquad\qquad\qquad\qquad\qquad\qquad\qquad$ □

435 The proof of Theorem 3.1 follows immediately from combining Lemmas A.4 and A.5.

# B Details from Section 4 (Experiments)

In this section, we give details from Section 4, and more experiments. First we describe the algorithms used in our experiments.

- Random Search consists of randomly choosing architectures and then training them, until the runtime budget is exceeded.
- Regularized evolution [14] consists of maintaining a population of neural architectures. In each iteration, a subset is selected and the best architecture from the subset is mutated. The mutation replaces the oldest architecture from the population. We used a population size of 30. We also found that replacing the worst architecture (not the oldest) performed better, so we used this version.
- Local search [25] is a simple greedy algorithm that has only recently been applied to NAS. We use the simplest instantiation (often called the hill-climbing algorithm).
- Bayesian optimization (BO) is a strong method for zeroth order optimization. We use the ProBO [12] implementation, which uses a Gaussian process kernel and expected improvement as the acquisition function.
- NASBOT [6] is a BO-based NAS algorithm. It was not defined for cell-based search spaces, so we use a variant that works for cell-based spaces [24].
- BANANAS [24] is a BO-based method which uses a neural predictor model.

## B.1 Experiments on NASBench-201

In this section, we give similar experiments to Figure 4.1, but with NASBench-201 instead of NASBench-101. Note that NASBench-201 is not as good for encoding experiments because every single architecture has the same graph structure - a clique of size 4. The only differences are the operations. Therefore, many encodings are functionally equivalent. For example, the one-hot, categorical, and continuous adjacency matrix encodings are all identical because the only difference is the way they encode the adjacency matrix. I.e., these encodings will all look like a set of operations, plus some adjacency matrix encoding that is the same for every architecture in the search space. The one-hot adjacency matrix encoding, path encoding, and truncated path encoding are all distinct from one another, so we run experiments with these encodings. See Figure B.1. We see largely the same trends as in NASBench-101 (Figure 4.1). Note that on the ImageNet-16-120 dataset, some algorithms such as NASBOT overfit to the training set, causing performance to decline over time.

## B.2 Best practices for NAS

Many authors have called for improving the reproducibility and fairness in experimental comparisons in NAS research [8, 27, 26], which has led to the release of a NAS best practices checklist [9]. We address each section and we encourage future work to do the same.

- **Best practices for releasing code.** We included our code in the supplementary material. We will release our code publicly after this reviewing cycle. We used the NASBench-101 and NASBench-201 datasets, so questions about training pipeline, evaluation, and hyperparameters for the final evaluation do not apply.
- **Best practices for comparing NAS methods.** We made fair comparisons due to our use of NASBench-101 and NASBench-201. We did run ablation studies and ran random search. We performed 300 trials of each experiment on NASBench-101 and NASBench-201.
- **Best practices for reporting important details.** We used the hyperparamters straight from the open source repositories, with a few exceptions listed above.

Figure B.1: Experiments on NASBench-201 with different encodings, keeping all but one subroutine fixed: *perturb architecture* (Reg. evolution (top row), local search (second row)), *train predictor model* (BANANAS (third row), Bayesian optimization (bottom row)).