[Reviews · NeurIPS 2020]

Review 1

Summary and Contributions: In the work, they do a thorough analysis and the several different architecture encodings for NAS. They characterize the scalability for each encoding and theoretical properties and empirically verify their theoretical findings. They then do a thorough experimental study of using different encodings with different NAS algorithms to disentangle the contribution of new encodings and algorithms in prior work.

Strengths: The work presents novel theoretical analysis of the scalability of NAS encodings and supports it with empirical evidence. It provides a detailed empirical study that is useful in disentangling encoding and algorithmic contributions of prior work using NASBench-101 and conducted a short experiment on generalizing outside the search space. This is highly relevant and both shows that more care needs to taken in choosing encodings for NAS and provides a strong results on the best encoding to use for several subroutines in NAS. They intend to release the code and included the code in the submission.

Weaknesses: A concern about the experiments comparing embeddings may be that though they used the hyperparameters and code released from previous work, those parameters may be tuned towards the particular embeddings that were used in previous work and could be biasing the results.

Correctness: The experimental methods are correct and support the claims. The only concern may be that the experiments may be strengthened by doing some hyperparameter sweeps for the different algorithms.

Clarity: Paper is well organized and clear. It might be beneficial to include more discussion of the results of the encoding experiments.

Relation to Prior Work: The work is well position to prior work.

Reproducibility: Yes

Additional Feedback: ================================== Post-Rebuttal I thank the authors for addressing my concerns in their rebuttal. The additional results with hyperparameter search learned encodings, and in the DARTS search space makes the paper much stronger. I increase my score (7->8) and recommend acceptance of the paper. ================================ It might be a bit easier to follow if the experiments to support the claims in section 3.1 came right after instead of immediately presenting the full study of architecture encodings.


Review 2

Summary and Contributions: This work presents the first formal study on the effect of architecture encodings for NAS, including a theoretical grounding and an empirical study. The authors(s) theoretically characterize the scalability of each encoding by quantifying the information loss from truncation. This study also provides an experimental study of architecture encodings for NAS algorithms and recommendations for best encodings to use in different settings.

Strengths: - Overall, this is a good empirical study on architecture encoding for NAS. The empirical and theoretical analysis of existing path-based and adjacency matrix based encoding is comprehensive. This study will provide inspirations to future works on encoding based NAS methods. -The experiments provide an ablation study for prior work by disentangling the algorithmic contributions from the encoding-based counterparts. This is helpful for evaluation the importance of architecture representation.

Weaknesses: - This is a good empirical study, yet lacking of new proposals for neural architecture encoding. A good empirical analysis not only provides insightful and inspirational observations or conclusions, but also contains new initial proposals based upon the analysis. - My major concern is the transferability of the architecture encodings. The author(s) should provide the experiments on real search spaces, such as DARTS space and the comment used MobileNetV3 space, and use the encoding methods for performance prediction. Performing comparisons with SOTA NAS approaches to verify the transferability and robustness of encoding based predictor. Although the author(s) provide the experiments on benchmarks, yet as we know, the search space of existing benchmarks are extremely small. The results on Benches are good, yet not sufficient enough for practical scenarios.

Correctness: This is an empirical study, the presentation of the experiments are sound.

Clarity: The writing of this paper is good, well-organized and easy to follow. The equations are clear.

Relation to Prior Work: The discussion with prior work is well-rounded. However, to prove the encoding-based NAS methods, we would like to see the results on more challenging and larger search spaces and comparisons with other NAS methods.

Reproducibility: Yes

Additional Feedback: =========== Post Rebuttal =========== The authors provide a preliminary evaluation on DARTS space, and will try their best to extend the experiments to a more full set of results in the final version of the paper. I do believe the authors will update the results. I lean to accept this paper. ==================================


Review 3

Summary and Contributions: =========== Post Rebuttal =========== I thank the authors for taking the time to address my review and conducting more experiments. With the new experiments the paper became certainly stronger. Also apologies that I missed the additional experiments on Nasbench201 in the appendix. I increase my score (6->7) and recommend acceptance of the paper. ================================= The paper studies the impact of various types of adjacency matrix and path encodings for neural network architectures, both theoretically and practically, and their effect on common sub-tasks of neural architecture search methods: random sampling, perturbation and training a predictor model. On the NASBench101 dataset the paper shows that the type of encoding indeed substantially effects performance of popular NAS methods, such as random search, regularized evolution or Bayesian optimization.

Strengths: The paper investigates the effect of the encoding of an architecture, which is arguably a rather under represented topic in the NAS literature, and shows that it is indeed an important design choice for many NAS approaches. On the long run, I could see that this paper raises awareness in the NAS community to be conscious about the influence of the encoding and maybe trigger the development of new encoding strategies that overcome current flaws.

Weaknesses: I think the main weakness of the paper is that it only considers NASBench101, i.e one type of search space and dataset. Unfortunately, while the results are interesting, it is not clear how the conclusions of the paper generalize to other search spaces and datasets. I do understand the computation requirements of such a study are non negligible but at least other tabular benchmarks, such as NASbench201 (even though it's search space is even smaller) should be included.

Correctness: Apart from only considering a single dataset, the claims and overall empirical evaluation seems to be sound.

Clarity: The paper is well written and easy to follow. Just a few things should be clarified: - Why does it say "We run at least 200 trials for each algorithm." in line 247? Conclusions would be more reliable if each method is run for the exact amount of trials. - Maybe you should consider to plot the log regret on the y-axis in Figure 4.1 to visualize the distance to the global optimum. - What is the computational overhead of the path encoding?

Relation to Prior Work: As far as I can tell the paper relates to all important prior work. The only nitpick that I have is that in Section 4 it says: "....continuous adjacency encoding has previously never been used for NAS with BO ...". I don't think it's correct. In NASBench101 they used this encoding for GP-based BO in Vizier as well as for BOHB.

Reproducibility: Yes

Additional Feedback: The paper argues that learned encoding, based on for example VAE, are part of the search strategy. I think this is a matter of perspective and, as the results indicate, would argue that search strategies in general depend on the encoding and vice versa. Because of that, I am convinced that the paper would become stronger if learned encoding, such as VAE or GraphNN, would be included in the comparison to see whether they overcome the issues of adjacency matrix or path encodings.

[Author Response · NeurIPS 2020]

We thank the reviewers for their helpful reviews. We performed all the additional experiments that were suggested: hyperparameter tuning for each individual encoding, (preliminary) experiments on the DARTS search space, and comparison to GraphNN's and VAE's (and we point out to R3 that experiments on NAS-Bench-201 were already in our paper). Please see the details below.

**R1**   We agree that it would strengthen our experiments to perform hyperparameter tuning on each NAS algorithm for each encoding. We just need to be careful that we do not perform hyperparameter tuning for specific *datasets* (in accordance with NAS best practices (Lindauer and Hutter, 2019; Yang et al. 2020)). Therefore, we perform the hyperparameter search on CIFAR-100 from NAS-Bench-201, and apply the results on NAS-Bench-101. We defined a search region for each hyperparameter of each algorithm, and then for each encoding, we ran 50 iterations of random search on the full hyperparameter space. We choose the configuration that minimizes the validation loss of the NAS algorithm after 200 queries. See the figure below for the results of Reg. Evolution (top left), and Local Search (top middle). Most encodings improved or stayed the same, though a few did slightly worse (because the hyperparameters for those encodings did not generalize from NAS-Bench-201 to NAS-Bench-101). Finally, we agree with the other comments/clarifications and will include them in the final version of the paper.

**R2**   While we do not have access to a large GPU cluster, we agree that results on the DARTS search space would strengthen the paper. We now provide preliminary results for experiments on the DARTS search space. Before this project, we had already computed a set of 1200 architectures from the DARTS search space trained to 50 epochs, which we now use to test different encodings on a neural predictor. The experimental setup is the same as in our paper, except the architectures were not all drawn i.i.d.—120 were drawn i.i.d. and the rest are mutations. We see similar trends as with the nas-bench-101/201 datasets. See the figure below (top right). Next, we ran an initial experiment testing three different encodings with random search. For each encoding, we trained 100 architectures to 25 epochs. See the figure below (bottom left). Here, the path-based encodings outperformed the adjacency encoding. With more time, we will train the architectures to more epochs. We also did not have time to run multiple trials, so there are no error bars. We will do our best to extend these to a more full set of results for the final version of the paper.

**R3**   We point out that our paper does include experiments from NAS-Bench-201 (briefly mentioned on lines 247-248, with the details in the appendix). We agree that we should include plots of the regret, and so we add an example below (bottom middle). We will correct all the other clarifications/suggestions you have mentioned. Finally, we agree that our paper would become stronger if we include learned encodings such as GraphNN and VAE. We found an open-source implementation for "Neural Predictor for Neural Architecture Search" and we also used the open-source code from the paper "D-VAE: A Variational Autoencoder for Directed Acyclic Graphs" (which was designed for the ENAS search space). We ran new experiments with these encodings in the same setting as our neural predictor experiment in our submission, and we show the new results in the figure below (bottom right). In fact, the path encoding outperforms the trainable encodings, which was also noticed in prior work, e.g. BANANAS (White et al. 2019). We will add these results to the paper.



[Meta-Review · NeurIPS 2020]

This paper thoroughly studies both the empirical and theoretical aspects of what encodings to use for representing architectures for the task of predicting their downstream final performance. This step is fundamental to many NAS pipelines. This is a fundamental contribution to NAS literature and should become the go-to paper to read for others trying to design their own NAS pipeline. The authors are encouraged to incorporate all reviewer comments to further improve their paper.